# Accuracy of Multimodality Fetal Imaging (US, MRI, and CT) for Congenital Musculoskeletal Anomalies

**DOI:** 10.3390/children10061015

**Published:** 2023-06-05

**Authors:** Roy U. Bisht, Mohan V. Belthur, Ian M. Singleton, Luis F. Goncalves

**Affiliations:** 1University of Arizona College of Medicine—Phoenix, 475 N. 5th St., Phoenix, AZ 85004, USA; 2Department of Child Health & Orthopedics, University of Arizona College of Medicine—Phoenix, 1919 E. Thomas Rd., Phoenix, AZ 85004, USA; 3Department of Surgery, Medical School, Creighton University, Omaha, NE 68178, USA; 4Department of Radiology, Phoenix Children’s Hospital, 1919 E. Thomas Rd., Phoenix, AZ 85016, USA; 5Child Health and Radiology, University of Arizona College of Medicine—Phoenix, 475 N. 5th St., Phoenix, AZ 85004, USA; 6Radiology, Mayo Clinic, 5777 E. Mayo Blvd., Phoenix, AZ 85054, USA

**Keywords:** fetal ultrasound, fetal MRI, low-dose computerized tomography, prenatal diagnosis, skeletal dysplasias

## Abstract

Background: Ultrasonography (US) is the first-line diagnostic tool used to assess fetal musculoskeletal (MSK) anomalies. Associated anomalies in other organ systems may benefit from evaluation via Magnetic Resonance Imaging (MRI). In this study, we compared the diagnostic accuracy of US and MRI to diagnose fetal MSK (primary objective) and non-MSK anomalies (secondary objective). We describe additional findings by low-dose computerized tomography (CT) in two cases incompletely characterized via US and MRI. Materials and Methods: This was an IRB-approved retrospective study of consecutive patients with suspected fetal MSK anomalies examined between December 2015 and June 2020. We compared individual MSK and non-MSK anomalies identified via US, MRI, and CT with postnatal outcomes. Sensitivity and specificity for US and MRI were calculated and compared. Results: A total of 31 patients with 112 MSK and 43 non-MSK anomalies were included. The sensitivity of MRI and US for MSK anomalies was not significantly different (76.6% vs. 61.3%, *p* = 0.3). Low-dose CT identified eight additional skeletal anomalies. MRI diagnosed a higher number of non-MSK anomalies compared to US (81.4% vs. 37.2%, *p* < 0.05). Conclusions: Fetal MRI and US have comparable sensitivity for MSK anomalies. In selected cases, low-dose CT may provide additional information. Fetal MRI detected a larger number of non-MSK anomalies in other organ systems compared to US. Multimodality imaging combining all the information provided by MRI, US, and CT, if necessary, ultimately achieved a sensitivity of 89.2% (95% CI: 83.4% to 95.0%) for the diagnosis of musculoskeletal anomalies and 81.4% for additional anomalies in other organs and systems.

## 1. Introduction

The incidence of fetal musculoskeletal (MSK) anomalies in pregnancy is approximately 0.4 to 0.6%, dropping to 0.024% postnatally, reflecting a high mortality rate [1]. Commonly identified fetal MSK anomalies include clubfeet, polydactyly, syndactyly, spinal deformities, limb-length discrepancies, skeletal dysplasias, and arthrogryposis [2,3,4,5,6]. Skeletal dysplasias are heritable diseases that affect bone and cartilage and occur in roughly 1/5000 births. It is of the utmost importance to properly diagnose skeletal dysplasias as early as possible in utero, as numerous are lethal [7]. Certain skeletal dysplasias carry a high risk of recurrence in future pregnancies, depending on the particular inheritance pattern [8]. For this reason, understanding the correct diagnosis can assist families in planning future pregnancies. Educating parents on the nature of the disease, the survival chances of the fetus, subsequent development abnormalities for survivors, and future reproductive risks are essential.

US is the primary imaging modality used to assess for congenital anomalies given its low cost, safety, ease of use, and availability [2]. Previously reported sensitivities for the prenatal diagnosis of skeletal dysplasias ranged from 53% to 67.9% [9,10,11,12]. A sensitivity of 63% has been reported for the prenatal diagnosis of clubfoot using US [9]. Regarding additional limb abnormalities, Dicke et al. found that US had a sensitivity of 19.1% prenatally for polydactyly, 76.0% for abnormal hand position, 76.0% for limb reduction defects, and 81.3% for arthrogryposis [10]. Data on the accuracy of fetal MRI in diagnosing MSK anomalies is limited with no direct comparison of the diagnostic accuracy between US and MRI [13,14]. Besides the evaluation of MSK anomalies per se, fetal MRI also has the potential to provide additional information in cases of syndromic skeletal dysplasias by diagnosing unsuspected associated anomalies in other organ systems (e.g., brain, lungs, kidneys, and GI tract) [2,11,12]. More recently, low-dose fetal computerized tomography (CT) with the three-dimensional (3D) reconstruction of the fetal skeleton has emerged as an attractive imaging modality for the accurate characterization of the skeletal phenotype in skeletal dysplasias [15,16,17,18]. Prior studies have shown that both the sensitivity and specificity of low-dose CT are higher compared to US, with the only limitation being the small radiation exposure in utero (usually <5 mSv) [15]. Thus, its use is limited to situations when US and/or MRI cannot satisfactorily characterize the phenotype [15].

At our institution, we perform approximately 250 fetal imaging evaluations per year using a combination of MRI and US. Anomalies referred for fetal MRI tend to be complex, usually identified via US at the obstetrician’s office and further evaluated via detailed US by maternal–fetal medicine specialists in the state of Arizona. Patients are referred for fetal MRI when the fetal phenotype has not been completely characterized. After review of the MRI images, the evaluation may be complemented by a targeted US and, in the case of skeletal anomalies, by a low-dose CT with 3D rendering of the fetal skeleton, but only in cases where the phenotype could not be characterized by fetal MRI or US.

The primary objective of this study is to determine if fetal MRI can provide additional diagnostic information compared to US for the evaluation of fetuses with a suspected MSK anomaly. The secondary objective is to determine if fetal MRI provides additional diagnostic information for anomalies not involving the MSK system in this group of fetuses (non-MSK anomalies). We describe additional findings identified with low-dose computerized tomography (CT) in cases incompletely characterized by US and MRI.

## 2. Methods

This was a retrospective IRB-approved study that included consecutive pregnancies with suspected fetal MSK anomalies referred to our institution for multimodality fetal imaging (fetal MRI, US, and low-dose CT if necessary) between December 2015 and June 2020. For each case, the mother’s prenatal chart, all images, and the infant’s postnatal chart were reviewed. Cases of intrauterine fetal demise without postmortem X-rays or autopsy and patients who were lost to follow up were excluded. For each case, the anomalies identified with the referring detailed prenatal US and the anomalies identified via each imaging modality performed at our institution were compared to postnatal diagnoses.

Fetal US was performed using an EPIQ Elite ultrasound system (Philips Healthcare, Bothell, WA, USA). Fetal MRI was performed using either a 3 Tesla Philips Ingenia MRI System or a 1.5 Tesla Philips Achieva MRI system (Philips Healthcare, Cambridge, MA, USA). Low-dose CT was performed using a 256-slice CT scanner (Philips 256-slice Brilliance iCT scanner, Philips Healthcare, Cambridge, MA, USA).

Most examined fetuses had more than one anomaly, and each anomaly was categorized as MSK or non-MSK. MSK anomalies were categorized as anomalies affecting the craniofacial structures, spine, clavicles, scapulae, ribs, pelvis, upper extremities, and lower extremities. Non-MSK anomalies were further categorized into cardiac, central nervous system, eye, gastrointestinal, or genitourinary anomalies. A unifying diagnosis based on phenotype was attempted and compared to a postnatal diagnosis established by postnatal clinical, imaging, and/or surgical evaluations for associations or sequences (e.g., amniotic band syndrome, caudal regression, or VACTERL), or postnatal clinical, imaging, and/or surgical and genetic testing for skeletal dysplasias or genetic anomalies (e.g., hypochondrogenesis or diastrophic dysplasia).

For statistical analysis, each individual anomaly was documented as either a true-positive or a false-positive diagnosis. This was performed for each separate modality (i.e., US, MRI, and CT) and compared to each individual anomaly diagnosed postnatally. For each system considered normal (i.e., CNS, cardiac, gastrointestinal, and genitourinary for non-MSK anomalies; cranial, facial, spine, clavicles, scapulae, ribs, pelvis, upper extremities, and lower extremities for MSK anomalies), either a true-negative or false-negative diagnosis was assigned after comparison with the postnatal outcome. Sensitivity and specificity with a 95% confidence interval (95% CI) were calculated for US and MRI and compared using McNemar’s test. The added value of CT, if any, is described separately. All *p*-values are two-sided, and *p* < 0.05 was considered statistically significant. Patient demographics and clinical characteristics are reported as means ± standard deviations for continuous variables and frequencies or percentages for categorical variables. Gestational age at the time of multimodality imaging was recorded.

## 3. Results

Forty consecutive singleton pregnancies with a suspected diagnosis of one or more fetal MSK anomalies were referred to our institution during the study period. Nine pregnancies complicated by intrauterine fetal demise without postmortem X-rays or autopsy (*n* = 4) and neonates who were lost to follow-up (*n* = 5) were excluded. Patient demographics are presented in Table 1. A total of 31 patients with 111 MSK anomalies and 43 non-MSK anomalies were included in this study. There were also 222 normal MSK findings and 112 normal non-MSK findings. All 31 patients underwent a fetal MRI. Twenty-one were further evaluated by a targeted US. Low-dose CT was performed in two cases (hypochondrogenesis and disorder of glycosylation mimicking Desbuquois dysplasia) [17]. Of the 31 patients included, four patients also genetic evaluation.

### 3.1. Diagnostic Accuracy for MSK Anomalies

Regarding MSK anomalies, the sensitivity of the referral US compared to postnatal outcome (*n* = 31) was 61.3% (95% CI: 52.5% to 70.3%). The sensitivity of MRI for the same cases was 76.6% (95% CI: 68.7% to 84.5%), but the difference was not statistically significant (McNemar’s test 10.7, *p* = 0.30) (Table 2).

In a sub-analysis restricted to cases that had matched US and MRI performed at our institution (*n* = 21), the sensitivity of US was 79.1% (95% CI: 70.5% to 87.7%) and the sensitivity for fetal MRI was 74.4%% (95% CI 65.2% to 83.6%), also not statistically significant (McNemar’s test 0.45, *p* = 0.50) (Table 3). When findings from US and MRI were combined, the sensitivity increased to 82.6% (95% CI: 74.5% to 90.6%), as US detected seven anomalies that were not identifiable by MRI (“cobrahead” appearance of the spine, mesomelic limb shortening of the upper and lower extremities in a case of diastrophic dysplasia, premature ossification center of the proximal femoral epiphysis, visualized an amniotic band that was not detectable by MRI in two fetuses, and correctly identified tibial hemimelia that could not be well visualized by MRI), whereas MRI identified three anomalies that were not identified by US (cleft palate, glossoptosis, and bell-shaped thorax).

Specificity was high for all methods: 94.6% (95% CI: 91.7% to 97.6%) for the referral US, 98.0% (95% CI: 95.7% to 99.9%) for US performed at our institution, 98.6% (95% CI: 97.1% to 99.9%) for fetal MRI, and 98.6% by a combination of US and MRI (95% CI: 97.7% to 99.6%).

### 3.2. Additional Skeletal Anomalies Diagnosed by Low-Dose CT

Eight additional osseous anomalies in two cases were identified only by low-dose CT. The first four were platyspondyly, round iliac wings with horizonatal acetabular roofs, demineralized sacrum, and metaphyseal flaring of the humeri in a fetus with hypochondrogenesis. The other four anomalies were enlarged sutures and fontanelles, coronal and sagittal clefts in the thoracolumbar spine, flat acetabula, and enlarged lesser trochanters of the femora (“sweedish key” or “monkey wrench sign”) in a case of a disorder of glycosylation mimicking Desbuquois dysplasia.

The combination of the information provided by US, MRI, and CT reached a sensitivity of 89.2% (95% CI: 83.4% to 95.0%), with a specificity of 98.2% (95% CI: 97.3% to 99.1%).

### 3.3. Diagnostic Accuracy for Non-MSK Anomalies

Regarding non-MSK anomalies (Table 4 and Table 5), the sensitivity was 37.2% (95% CI: 22.8% to 51.7%%) for the referral US. The sensitivity of US performed at our institution was 48.1% (95% CI: 29.3% to 67.0%). The sensitivity increased to 81.4% (95% CI: 69.8% to 93.0%) for fetal MRI (McNemar’s test 7.5, *p* < 0.05 for the comparison between fetal MRI and referral ultrasound; McNemar’s test 2.77, *p* = 0.10 for the comparison between fetal MRI and ultrasound performed at our institution). Specifically, MRI added information in 4/31 cases by correctly identifying 10 additional anomalies, most of which affected prognosis [imperforate anus (*n* = 1), malformations of cortical development (*n* = 4), cerebellar vermis hypoplasia/dysplasia (*n* = 2), agenesis of the corpus callosum (*n* = 1), microphthalmia (*n* = 1), and coloboma (*n* = 1)].

The specificity was 95.5% (95% CI: 91.7–99.4%) for the referring US, 97.4% (95% CI: 93.8% to 99.9%) for the US performed at our institution, and 98.2% (95% CI: 95.8% to 99.9%) for fetal MRI.

A detailed list of anomalies and whether they were identified by US, MRI, or CT is shown in Table 6.

## 4. Discussion

This study showed that US and MRI have comparable sensitivities for the prenatal diagnosis of MSK anomalies and that, in a population of fetuses with a skeletal anomaly, MRI may add information by the identification of previously unsuspected anomalies affecting other organs and systems. The number of cases evaluated by low-dose CT in this study is small (*n* = 2), but in these two cases, CT identified additional skeletal anomalies that helped achieve an accurate prenatal characterization of the phenotype in a case of hypochondrogenesis and a case of a disorder of glycosilation mimicking Desbuquois dysplasia. While CT was not used frequently, incorporating this modality can provide a more detailed assessment of the fetal skeleton when compared to both US or MRI and can, as a result, allow for a more definitive diagnosis [18].

Doray et al. have previously studied the efficacy of US to prenatally identify skeletal dysplasias [19]. There were 47 cases with skeletal dysplasia that were identified with the prenatal and postnatal diagnoses compared. Of the 47 cases, 28 (60%) had an accurate prenatal diagnosis using ultrasonography [19]. This was similar to the results found in the current study, where the referring USs had a sensitivity of 58.9% for all MSK anomalies, not limited to skeletal dysplasia. Of the remaining cases, 9 (19%) had an inaccurate diagnosis, and 10 (21%) had an imprecise diagnosis [19]. Similarly, Parilla et al. examined the prenatal accuracy of US in diagnosing skeletal dysplasia over eight years [20]. In the 31 cases examined in that study, 20 (65%) had an accurate prenatal diagnosis using US. Of note, lethality was correctly predicted in 16 out of 16 eligible cases (100%) [20]. This finding was also seen by Goncalves et al., who found that US had a sensitivity of 89% in prenatally identifying a lethal dysplasia [21]. While the overall diagnosis was not always accurate, US was able to correctly predict lethality when applicable. US is an incredibly valuable tool in prenatal imaging; however, the findings in the studies by Doray et al. and Parilla et al. in addition to this current study suggest that the use of US leaves significant room for improvement in the diagnosis of MSK anomalies and, particularly, better characterization of anomalies in other involved organs and systems.

At the moment, there is limited research regarding the accuracy of stand-alone MRI for congenital MSK anomalies. Blaicher et al. examined the utility of fetal MRI in 14 patients that were found to have skeletal dysplasia in prenatal US [22]. In ten of those cases, US was more accurate in diagnosing skeletal dysplasia than MRI. In the other four cases, each with spina bifida, MRI provided additional information that was beneficial in presurgical planning [22]. Studies that featured additional organ systems showed different results. Goncalves et al. found that when examining central nervous system (CNS) anomalies prenatally, MRI was more sensitive than both 3D US and 2D US, 88.9% compared to 66.7% and 72.2%, respectively [23]. These results for CNS anomalies were similar to the ones seen in this current study, where US had a sensitivity of 26.9% for CNS anomalies, while multimodality imaging had a sensitivity of 96.2%. The same study by Goncalves et al. showed that when MRI alone was compared to 3D US and 2D US for non-CNS anomalies, the sensitivities for each modality were similar [23]. MRI also has demonstrated utility in differentiating isolated versus complex anomalies, such as amniotic band syndrome in a case of isolated limb deficiency [24]. This added diagnostic value not only allows providers to adequately approach a child’s treatment but also allows the family to fully comprehend the complexity of the congenital anomalies.

One of the limitations of this study was that it was a retrospective chart review and allowed us to exclude patients who did not qualify for this study. This was also a single-institution study that limited the patient population. MRI was used more frequently than US, and there were only two cases where CT was used in this study, so future studies could aim at comparing a more equal number of cases from each modality. The majority of MSK anomalies were seen in the extremities, so future studies could include patients with MSK findings localized to other parts of the body. Of the 31 patients included, only four had a follow-up genetic analysis, so there was limited correlating genetic data for many of the cases.

Diagnosing MSK anomalies and skeletal dysplasias accurately in the prenatal setting is of the utmost importance. Given the morbidity and mortality associated with certain severe skeletal dysplasias, it is essential to educate parents on the disease so they can prepare for potentially unfavorable outcomes. While US and MRI demonstrated similar diagnostic accuracy to diagnose MSK anomalies, the use of MRI provided a more accurate assessment for non-MSK anomalies. Multimodality imaging combining all the information provided by MRI, US, and CT if necessary ultimately achieved a sensitivity of 89.2% (95% CI: 83.4% to 95.0%) for the diagnosis of musculoskeletal anomalies and 81.4% for additional anomalies in other organs and systems.

## Figures and Tables

**Table 1 children-10-01015-t001:** Patient Demographics.

Gestational Age at Prenatal Diagnosis (Weeks) Mean (S.D.)	28.7 (4.8)
**Ethnicity**	
White/Caucasian	51.6% (16/31)
Hispanic	22.6% (7/31)
Native American	6.4% (2/31)
Asian	3.2% (1/31)
Black/African American	3.2% (1/31)
Other	3.2% (1/31)
Unknown	9.7% (3/31)
**Fetal gender**	
Male	58.1% (18/31)
Female	41.9% (13/31)

**Table 2 children-10-01015-t002:** Accuracy of Referral US vs. Fetal MRI for MSK anomalies (31 patients and 111 individual anomalies).

	TP	FN	Sensitivity	TN	FP	Specificity
Referral Ultrasound	68	43	61.3%	210	12	94.6%
fetal MRI	85	26	76.6%	219	3	98.6%

McNemar’s test 1.06, *p* = 0.3. TP: true-positive diagnosis. FN: false-negative diagnosis. TN: true-negative diagnosis. FP: false-positive diagnosis.

**Table 3 children-10-01015-t003:** Accuracy of US at our institution vs. fetal MRI for MSK anomalies (21 patients and 86 individual anomalies).

	TP	FN	Sensitivity	TN	FP	Specificity
Ultrasound	68	18	79.1%	145	3	98.0%
Fetal MRI	64	22	74.4%	146	2	98.6%

McNemar’s test 0.45, *p* = 0.50. TP: true-positive diagnosis. FN: false-negative diagnosis. TN: true-negative diagnosis. FP: false-positive diagnosis.

**Table 4 children-10-01015-t004:** Accuracy of Referral US vs. Fetal MRI for non-MSK anomalies (31 patients and 43 individual anomalies).

	TP	FN	Sensitivity	TN	FP	Specificity
Referral Ultrasound	16	27	37.2%	5	107	95.5%
Fetal MRI	35	8	81.4%	112	2	98.2%

McNemar’s test 7.5, *p* < 0.001. TP: true-positive diagnosis. FN: false-negative diagnosis. TN: true-negative diagnosis. FP: false-positive diagnosis.

**Table 5 children-10-01015-t005:** Accuracy of US at our institution vs. fetal MRI for non-MSK anomalies (21 patients and 27 individual anomalies).

	TP	FN	Sensitivity	TN	FP	Specificity
Ultrasound	13	14	48.1%	75	2	97.4%
Fetal MRI	20	7	74.1%	75	2	96.4%

McNemar’s test 2.77, *p* = 0.10. TP: true-positive diagnosis. FN: false-negative diagnosis. TN: true-negative diagnosis. FP: false-positive diagnosis.

**Table 6 children-10-01015-t006:** Musculoskeletal anomalies and non-musculoskeletal anomalies diagnosed postnatally in patients that underwent multimodality imaging.

Case Number and Gestational Age at Diagnosis	Postnatal Syndromic Dx	Anomalies at Referral US	Additional Anomalies MRI	Additional Anomalies by US (at Our Institution)	Additional Anomalies CT	Anomalies Confirmed	Anomalies Missed or Syndromic Dx Not Made	False-Positive Diagnoses	Additional Information from MMI Compared to Referral US
2 32w0d	Amniotic band sequence	Amniotic band/soft tissue constriction Clubhand Forearm shortening Proposed syndromic Dx: amniotic band sequence	None	Not performed	Not performed	Amniotic band/soft tissue constriction Club hand Forearm shortening Proposed syndromic Dx: amniotic band sequence	Referral US and MRI Fractured humerus	None	None
3 31w3d	No syndromic dx	Micrognathia Short long bones	None	Not performed	Not performed	None	None	Referral US: Micrognathia Short long bones	None
4 28w5d	Amniotic band sequence	Absent right hand	Forearm amputation (includes absent right hand) Amniotic band Proposed syndromic Dx: amniotic band sequence	Not performed	Not performed	Forearm amputation (includes absent right hand)	Referral US: Forearm amputation Amniotic band	None	Forearm amputation Amniotic band
5 30w3d	No syndromic dx	Craniosynostosis	Hypertelorism Midface hypoplasia	Not performed	Not performed	Craniosynostosis Hypertelorism Midface hypoplasia	Referral US: Hypertelorism Midface hypoplasia	None	Hypertelorism Midface hypoplasia
6 31w2d	No syndromic dx	Butterfly vertebrae Clubfoot Fused vertebrae Hemivertebrae	Blunt conus medullaris	Not performed	Not performed	Butterfly vertebra Blunt conus medullaris Clubfoot Congenital vertical talus Fused vertebrae Hemivertebrae	Referral US: Blunt conus medullaris Referral US and MRI: Congenital vertical talus	None	Blunt conus medullaris
7 20w0d	Caudal regression sequence	Lower limb contractures Lumbar and sacral agenesis Proposed syndrome Dx: caudal regression sequence	Blunt conus medullaris Proposed syndrome Dx: caudal regression sequence	Not performed	Not performed	Blunt conus medullaris Lower limb contractures Lumbar and sacral agenesis VSD	Referral US: Blunt conus medullaris MRI: VSD	None	Blunt conus medullaris
8 36w3d	Arthrogryposis multiplex congenita with normal whole exome sequencing and mitochondrial genome testing	Micrognathia Lower/upper limb contractures Proposed syndromic Dx: Arthrogryposis	High arched palate	Not performed	Not performed	High-arched palate Micrognathia Lower/upper limb contractures	Referral US: High-arched palate	None	High-arched palate
9 25w3d	Diastrophic dysplasia (confirmed mutation in SLC6A2)	Abducted thumb Brachydactyly Clubfoot Hypertelorism Lordosis lumbosacral spine Mesomelic shortening of upper and lower extremities Micrognathia Midface hypoplasia Short ribs	Cleft palate Glossoptosis Proposed syndromic Dx: Diastrophic dysplasia	Cobrahead appearance on the spine	Not performed	Abducted thumb Bilateral hip dislocation Brachydactyly Cleft palate Clubfoot Cobrahead appearance on the spine Glossoptosis Lordosis lumbosacral spine Mesomelic shortening of upper and lower extremities Micrognathia	Referral US: Bilateral hip dislocation Cleft palate Cobrahead appearance on the spine Glossoptosis US and MRI: Bilateral hip dislocation	Referral US: Hypertelorism Midface hypoplasia Short ribs	Cleft palate Cobrahead appearance on the spine Glossoptosis
10 23w1d	VACTERL	Hemivertebrae Scoliosis Bilateral small pelvic kidneys Proposed syndromic dx: VACTERL	Abnormal ribs Proposed syndromic dx: VACTERL	None	Note performed	Hemivertebrae Left renal agenesis Scoliosis	Referral US: Abnormal ribs	Referral US: Small right pelvic kidney	Abnormal ribs
12 35w5d	Arthrogryposis	Hypotonic upper and lower extremities Skull indentation	Hypotonic upper and lower extremities Contractures in the upper and lower extremities Skull indentation is seen but attributed to mass effect from a maternal rib (normal) Proposed syndromic dx: arthrogryposis	None	Not performed	Contractures in upper and lower extremities Hypotonic upper and lower extremities	Referral US: Contractures in upper and lower extremities	Referral US: Skull deformity MMI: Skull deformity	Contractures in upper and lower extremities
13 35w6d	No syndromic dx	Clubfoot Small head	None	None	Not performed	Clubfoot	None	Referral US: Small head	None
14 36w0d	Proximal focal femoral deficiency	Short femurs	Bell-shaped thorax Proposed syndromic Dx: Asphyxiating thoracic dysplasia	Premature ossification of the proximal femoral epiphysis Proposed syndromic Dx: Asphyxiating thoracic dysplasia	Not performed	Bell-shaped thorax Premature ossification proximal femoral epiphysis Short femurs	Referral US: Bell-shaped thorax Premature ossification proximal femoral epiphysis Proximal focal femoral deficiency MMI: Proximal focal femoral deficiency	None	Bell-shaped thorax Premature ossification proximal femoral epiphysis
18 23w1d	Amniotic band sequence	Amniotic band right leg Left clubfoot Proposed syndromic Dx: amniotic band sequence	Pseudoarthrosis right tibia/fibula Proposed syndromic Dx: amniotic band sequence	Proposed syndromic Dx: amniotic band sequence	Not performed	Amniotic band on right leg Left clubfoot Pseudoarthrosis right tibia/fibula	Referral US: Pseudo-arthrosis right tibia/fibula	None	Pseudoarthrosis right tibia/fibula
19 23w6d	Hypochondrogenesis	Abnormal mineralization of the spine Clubfoot Micrognathia Micromelia of upper and lower extremities Small, bell-shaped thorax Proposed syndromic Dx: Hypochondrogenesis	None Proposed syndromic Dx: Hypochondrogenesis	None Proposed syndromic Dx: Hypochondrogenesis	Metaphyseal flaring of the humeri Platyspondyly Round iliac wings with horizonal acetabular roof Proposed syndromic Dx: Hypochondrogenesis	Abnormal mineralization of the spine Clubfoot Metaphyseal flaring of humeri Micromelia of upper and lower extremities Platyspondyly Small, bell-shaped thorax	Referral US: Metaphyseal flaring of humeri US and MRI at our institution: Metaphyseal flaring of the humeri Platyspondyly Round iliac wings with horizonal acetabular roof	Referral US: Micrognathia	Metaphyseal flaring of humeri Platyspondyly Round iliac wings with horizonal acetabular roof
21 28w0d	Caudal regression sequence	Clubfoot Hypoplastic lower extremities Lumbar and sacral agenesis Proposed syndromic Dx: caudal regression sequence	Horseshoe kidney Imperforate anus Proposed syndromic Dx: caudal regression sequence	None Proposed syndromic Dx: caudal regression sequence	Not performed	Clubfoot Horseshoe kidney Imperforate anus Hypoplastic lower extremities Lumbar and sacral agenesis	Referral US: Horseshoe kidney Imperforate anus	None	Horseshoe kidney Imperforate anus
22 26w6d	No syndromic dx	Right fibular hemimelia	None	None	Not performed	Four ray foot Right fibular hemimelia	Referral US: Four ray foot MRI and US at our institution: Four ray foot	None	None
23 20w4d	VACTERL	Bilateral hydronephrosis Interrupted IVC Left SVC Right clubfoot Right radial aplasia VSD Proposed syndromic Dx: VACTERL, trisomy 18	Spinal dysraphism (tiny defect at the terminus thecal sac) No VSD No proposed syndromic DX	None. No VSD No proposed syndromic DX	Not performed	Bilateral hydronephrosis Defect of terminus thecal sac Interrupted IVC Left SVC Right clubfoot Right radial aplasia	Referral US: Spinal dysraphism (tiny defect at the terminus thecal sac) Duodenal atresia Imperforate anus TEF MMI: Duodenal atresia Imperforate anus TEF No proposed syndromic Dx	None	Spinal dysraphism (tiny defect at the terminus thecal sac)
24 33w0d	Amniotic band sequence	Below knee amputation No proposed syndromic Dx	None	Amniotic band Proposed syndromic Dx: amniotic band sequence	Not performed	Amniotic band Below knee amputation	Referral US: Amniotic band MRI: amniotic band	None	Amniotic band
25 33w0d	Congenital disorder of glycosylation mimicking Desbuquois dysplasia	Clubfoot Midface hypoplasia Small, bell-shaped thorax No proposed syndromic Dx	Abnormal brain gyration and sulcation Cerebellar vermis hypoplasia Dysgenesis of the corpus collosum Enlarged lesser trochanters Enlarged sutures and fontanelles Incompletely rotated hippocampi Periventricular heterotopia Short first metacarpal Ventriculomegaly No proposed syndromic Dx	None	Coronal and sagittal clefts in the thoracolumbar spine Enlarged sutures and fontanelles Enlarged lesser femoral trochanters Flat acetabula Proposed syndromic Dx: Desbuquois dysplasia	Abnormal gyration and sulcation Cerebellar vermis hypoplasia Coronal and sagittal clefts in the thoracolumbar spine Clubfoot Dysgenesis of the corpus collosum Enlarged lesser femoral trochanters Enlarged sutures and fontanelles Flat acetabula Incompletely rotated hippocampi Midface hypoplasia Periventricular heterotopia Radial head dislocation Small, bell-shaped thorax Short first metacarpal Ventriculomegaly	Referral US: Abnormal gyration and sulcation Cerebellar vermis hypoplasia Cleft soft palate Dysgenesis of the corpus collosum Enlarged sutures and fontanelles Flat acetabula Incompletely rotated hippocampi Periventricular heterotopia Radial head dislocation Ventriculomegaly US at our institution: Cerebellar vermis hypoplasia Cleft soft palate Dysgenesis of the corpus collosum Enlarged sutures and fontanelles Flat acetabula Incompletely rotated hippocampi Periventricular heterotopia Radial head dislocation Ventriculomegaly MRI: Coronal and sagittal clefts in the thoracolumbar spine Enlarged sutures and fontanelles Enlarged lesser femoral trochanters Flat acetabula	US: Broad maxilla MRI: Ulnar deviation of fingers	Abnormal gyration and sulcation Cerebellar vermis hypoplasia Coronal and sagittal clefts in the thoracolumbar spine Dysgenesis of the corpus collosum Enlarged lesser trochanters Enlarged sutures and fontanelles Flat acetabula Incompletely rotated hippocampi Periventricular heterotopia Short first metacarpal Ventriculomegaly
26 32w6d	No syndromic dx	Bilateral clubfoot Tibial hemimelia No proposed syndromic Dx	Missing 1st digital ray right foot + syndactyly Missing two digital rays on left foot No proposed syndromic Dx	None	Not performed	Bilateral clubfoot Missing 1st digital ray right foot + syndactyly Missing two digital rays on left foot Tibial hemimelia	Referral US: Missing 1st digital ray right foot + syndactyly Missing two digital rays on left foot	None	Missing 1st digital ray right foot + syndactyly Missing two digital rays on left foot
27 29w5d	Amniotic band sequence	Abnormal left hand Bilateral clubfoot Possible amniotic band Proposed syndromic Dx: amniotic band sequence		Amniotic band of tissue on left forearm Proposed syndromic Dx: amniotic band sequence		Acrosyndactyly 2nd–4th fingers on left hand Amniotic band of tissue on left forearm Bilateral Clubfoot Bilateral great toe amputation	Referral US: Bilateral great toe amputation MMI: Bilateral great toe amputation	None	None
28 25w5d	No syndromic dx	Displaced right foot Hypoplastic R tibia/fibula concerning fibular/tibial hemimelia No proposed syndromic Dx	None No proposed syndromic Dx	None No proposed syndromic Dx	Not performed	Calcaneovalgus right foot deformity Congenital posteromedial tibia/fibula bowing w/secondary right shorter than left leg length discrepancy	None	None	None
29 27w6d	Abnormal CMA: gain of 75 Mb of DNA from chromosome 7 at band q21.11q36.3, including 387 OMIM genes of clinical significance	Bilateral ventriculomegaly Dilated pulmonary artery Dysplastic right kidney Prominent cisterna magna VSD No proposed syndromic Dx	Bell-shaped thorax Delayed myelination of parietal and occipital lobes Dysgenesis of corpus collosum No proposed syndromic Dx	None No proposed syndromic Dx	Not performed	Bell-shaped thorax Bilateral ventriculomegaly Delayed myelination of parietal and occipital lobes Dysplastic right kidney Dysgenesis of corpus collosum Prominent cisterna magna VSD	Referral US: Cleft palate Delayed myelination of parietal and occipital lobes Dysgenesis of corpus collosum Tethered cord US at our institution: Cleft palate Delayed myelination of parietal and occipital lobes Dysgenesis of corpus collosum Tethered cord MRI: Cleft palate VSD Tethered cord	Referral US Dilated pulmonary artery Flat facial profile Hypertelorism Tricuspid regurgitation US at our institution: Dilated pulmonary artery Abnormal posturing of upper and lower extremities MRI: Abnormal posturing of upper and lower extremities Hypoplastic cerebellum Hypoplastic kinked brainstem	Bell-shaped thorax Delayed myelination of parietal and occipital lobes Dysgenesis of corpus collosum
30 22w0d	Klippel-Feil Syndrome	Abnormal cervical spine No proposed syndromic Dx	Closed thoracic spinal dysraphism Fused ribs Sprengel’s deformity with omovertebral bone Unilateral renal agenesis Proposed syndromic Dx: Klippel-Feil syndrome	None.	Not performed	Closed thoracic spinal dysraphism Fused ribs Multiple segmentation abnormalities of the thoracic cervical spine Sprengel’s deformity with omovertebral bone Unilateral renal agenesis	Referral US: Closed thoracic spinal dysraphism Fused ribs Sprengel’s deformity with omovertebral bone Unilateral renal agenesis	None	Closed thoracic spinal dysraphism Fused ribs Sprengel’s deformity with omovertebral bone Unilateral renal agenesis
31 35w1d	No syndromic dx	Dandy-Walker malformation Ventriculomegaly No proposed syndromic Dx	ACC w/interhemispheric cyst Butterfly vertebrae Coloboma Gray matter heterotopia Microphthalmia Polymicrogyria Proposed syndromic Dx: Aicardi syndrome		Not performed	ACC w/inter-hemispheric cyst Butterfly vertebrae Coloboma Gray matter heterotopia Hypoplastic cerebellar vermis with rotation Microphthalmia Polymicrogyria Ventriculomegaly	Referral US: ACC w/inter-hemispheric cyst Butterfly vertebrae Coloboma Gray matter heterotopia Microphthalmia Polymicrogyria	None	ACC w/inter-hemispheric cyst Butterfly vertebrae Coloboma Gray matter heterotopia Microphthalmia Polymicrogyria
33 31w6d	Amniotic band sequence	Abnormal fetal hands with missing digits No proposed syndromic Dx	None.	Amniotic band	Not performed	Amniotic band Bilateral hand deformation (right-hand partial amputation of thumb, index, and middle fingers. Left hand—shortened and small thumb, index fingers, middle finger with circumferential indentation, webbed toes)	Referral US: Amniotic band MRI: amniotic band	None	Amniotic band
34 21w3d	Proximal focal femoral deficiency	Absent right tibia and fibula Bilateral bowed short femurs Cloverleaf skull Left clubfoot No proposed syndromic Dx	Right foot abnormally rotated Tiny dysmorphic R femur Proposed syndromic dx: proximal focal femoral deficiency	Right tibial hemimelia	Not performed	Left clubfoot Left bowed short femur Tiny dysmorphic right femur Right foot abnormally rotated Right tibial hemimelia	Referral US: Right foot abnormally rotated Right tibial hemimelia MRI: Right tibial hemimelia	Referral US: Cloverleaf skull	Right foot abnormally rotated
36 22w3d	No syndromic dx	Abnormal R 2nd–5th toes Absent right fibula Short bowed right tibia	None	None	Not performed	Abnormal R 2nd–5th toes Absent right fibula Short bowed right tibia	None	None	None
37 22w5d	VACTERL	Fixed contracture of the wrist Hemivertebrae Radial ray aplasia Right fingers poorly visualized Pyelectasis right kidney Proposed syndromic Dx: VACTERL	Absent first and second digital ray Proposed syndromic Dx: VACTERL	None	Not performed	Absent first and second digital ray Club hand Hydro-nephrosis Radial ray aplasia	Referral US: Absent first and second digital ray VSD MMI: VSD	None	Absent first and second digital ray
38 26w0d	Caudal regression sequence	Absent sacrum Left clubfoot Proposed syndromic DX: caudal regression sequence	Blunted conus at T12	None.	Not performed	Agenesis of lumbosacral spine beyond L5 Blunted conus at T12 Left clubfoot Piriform aperture stenosis VSD	Referral US: Blunted conus at T12 Piriform aperture stenosis VSD MMI: Piriform aperture stenosis VSD	None	Blunted conus at T12
40 35w6d	No syndromic dx	Bilateral Clubfoot Dysgenesis of corpus callosum Microcephaly Micrognathia Ventriculomegaly No proposed syndromic Dx	Contractures at knees and ankles Malformation of cortical development Schizencephaly	None.	Not performed	Bilateral Clubfoot Contractures at knees and ankles Dysgenesis of corpus callosum Malformations of cortical development Microcephaly Micrognathia Schizencephaly Ventriculomegaly	Referral US: Contractures at knees and ankles Malformation of cortical development Schizencephaly	None	Contractures at knees and ankles Malformation of cortical development Schizencephaly

ACC: agenesis of the corpus callosum. CT: computerized tomography. Dx: diagnosis. MMI: multimodality imaging (US, MRI + CT if necessary). MRI: Magnetic Resonance Imaging. TEF: tracheoesophageal atresia with fistula. US: ultrasound. VSD: ventricular septal defect.

## Data Availability

Data is described in detail on Table 6.

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
