# Peer review of "Accuracy of Multimodality Fetal Imaging (US, MRI, and CT) for Congenital Musculoskeletal Anomalies"

_children, 2023, doi:10.3390/children10061015_

Round 1

Reviewer 1 Report

I appreciate the authors’ effort to describe however the main problems is significant flaws of the study and this study add little to the existing knowledge in literature.

Abstract

Objective is not relevant heading, background is more suitable. Only the last sentence is objective.

Methods: please specify study design and define clearly referring US / targeted US. What is the true study group.

Results: non-musculo-skeletal anomaies (MSA) is not included in the objectives. This may included as secondary objectives.

Conclusion: Please more specify that MMI is better than what all referring US or US with uncertain diagnosis etc.

What is keywords?  Key words: m ?

Introduction: rationale is not valid. Ultrasound alone is compared to US + MRI + CT? The latter can be expected to predict better in accuracy but whether or not it is cost-effective is more important issue.

What is the rationale for MSA and non-MSA; why not CNS and non-CNS etc.

Please clearly indicate the main objective (accuracy in predict musculo-skeletal anomaies: MSA) and second objective (non-MSA)

Methods:

The study design is not clear. It seems to be diagnostic test study, which it has to have disease and non-disease cases for calculation. In results, the authors mention 223 normal cases (no disease) but do not mention in “Methods”, why / how this normal group recruited to undergo MMI. So the authors must clearly define what is the study population (? cases with referring US with suspected of MSM and all of them must also undergo MMI and use Postnatal diagnosis as a gold standard.).

In actual practice, all referring US underwent targeted US (repeated) at the authors’ center, right!. I understand that not all of them proceeded to MMI. Certainly, in some cases, repeated US could give definitive diagnosis and need no further MMI. Therefore, some cases underwent targeted US without MMI. How many cases. I suggest the authors create flow chart of recruitment. From total referring US, repeated US (targeted) in your center, to the remaining cases with uncertain diagnosis in need of MMI. Please clarify how they selected cases to undergo MMI (certainly not routine, right!)

It seems to me that referring US in this study is referring US which definite diagnosis could not be made by US. Exactly, this study did not compare the accuracy of all cases ultrasound (but only uncertain cases) with MMI. I am confused. So please clearly indicate whether or not all cases of referring US to this center underwent MMI or only some cases with uncertain diagnosis.

Please indicate that all consecutive cases of prenatal diagnosis of musculo-skeletal anomaies (MSA) were included.

The authors should specifically define: referring US, targeted US, performed by whom? The same team as that performed MMI. So, targeted US vs MMI may be better than referring US vs MMI.

Machine models used in the study should be specified.

Multimodality is heterogeneous/ not uniformly the same in all cases, depending on degree of difficulty in diagnosis. For example, low-dose CT was performed if necessary. Most cases they add targeted US to MMI, in spite of the fact that referring US (performed by MFM specialist) similar to targeted US (performed in authors’ center). I think that the study may be better if the authors compare targeted US (in their center) with MRI or MRI+targeted US vs targeted US alone. This will increase reliability of the results to address how much MRI can add to what targeted US can do.

Since you have two US exam types (referring and targeted as a part of MMI), do you use referring US or targeted US? You have to used targeted US alone and see if MMI add or not. Please clarify what is study population

Results:

It’s hard to understand, why 31 patients but 112 + 49 anomalies. This means that on patients had 4-5 anomalies!!!.

Why number of cases with MSA is much higher than non-MSA (!!!; please make a comment)

Only 2 case of CT included in this study. So the two cases should be excluded and change the study from comparing referring US and MMI (US+MRI) or to comparing referring US with MRI.

Discussion

You may be mention the following limitation to your “discussion”. “Not all cases done by MMI. Sens spec in this study not applied for general practice but they represent only difficult cases which could not be diagnosed by refereeing US.” Sensitivity of referring is certainly dependent on skill of sonographers who did referring US.

I believe that a majority of MSA could be diagnosed with simple US and not referred to MMI.”

Reviewer 2 Report

1. Small retrospective cohort

2. Can you give more rationale to the methodology; you get a referral with US you then go directly to MRI and if needed focused US and the low dose CT Total 31 / 31 MRI - 23 US  - 2 CT ; can you explain the reasons that you went to US and again to CT has this would help with the strength or weakness of the screen vs diagnosis.

3. Can you provide genetic diagnosis and  genomic exomes as part of your work-up?

4. Can you comment on patient related recurrence risks ?

5. Can you create your algorithm so others can consider this process?

Reviewer 3 Report

This article aligns with the other studies in the literature regarding the imaging investigation of congenital musculoskeletal anomalies. The present study shows the utility of quantifying the role of imaging in terms of adverse pregnancy outcomes in the case of these anomalies. The authors provided adequate details about the methodology, evaluation, findings, and investigations.

However, some suggestions could improve the quality of the article:

1.     Materials and Methods “between December 2015 and June 2020 and our institution.” Typo mistake

2.     Missing the keywords in the abstract section 

3.     In Table 1, the standard deviation of gestational age at prenatal diagnosis has been omitted

4.      “The same study by Goncalves et al. showed that when MRI alone was compared to 3D-ultrasound and 2D-ultrasound for non-CNS anomalies, the sensitivities for each modality were similar”. Please add reference

5.     How many cases had a genetic evaluation done?

6.     How often was the monitoring of these cases carried out?

7.     It is interesting to include in the table the gestational age at which the condition was first diagnosed because it can vary depending on which segment of the body is affected

8.     The role of dynamic MRI studies in temporary fetal malpositions

9.     For a better understanding, a classification should have been made according to the affected segment: cranial bone structures, thorax, spine, long bones, and muscular system. The discussions should have been systematized according to this topography.

10.  Prenatal diagnosis of musculoskeletal abnormalities should be based on imaging information and genetic studies.

11.  The role of fetal MRI in differentiating between isolated and complex anomalies.

12.  Application of 3D computerized tomography for in utero diagnosis of skeletal dysplasias after 30 weeks of pregnancy.

Round 2

Reviewer 1 Report

Comment Report 2

I really appreciate the authors’ effort to improve the manuscript, which has improved significantly. However, the main problem could not be fixed, concerning study design retrospective nature without clear inclusion criteria (not prospective standard protocol). The cases were recruited with subjective approach or less reliable from the referral hospitals. Also, the results could not make generalization, since they are derived from very specific group of cases. I suggest reject the manuscript or possibly accept with low priority.

Author Response

Thank you for the comments, we appreciate your review and assistance. 

Reviewer 2 Report

The responses to my original questions were only marginally answered but the added tables allowed your conclusion to be better focused.

Author Response

(The authors gave the same response as above.)

Reviewer 3 Report

Although the retrospective study had a small number of cases, this article aligns with efforts to achieve the best possible management of these cases. The authors tried to systematize these cases and made the changes according to the recommendations.

Kind regards

Author Response

(The authors gave the same response as above.)
